# A Si IGBT/SiC MOSFET Hybrid Isolated Bidirectional DC–DC Converter for Reducing Losses and Costs of DC Solid State Transformer

Jun Huang [1,*], Yu Wang [1], Zhenfeng Li [2], Hongbo Zhu [1] and Kai Li [1]

[1] State Key Laboratory of Reliability and Intelligence of Electrical Equipment, School of Electrical Engineering, Hebei University of Technology, Tianjin 300401, China; 202121401086@stu.hebut.edu.cn (Y.W.); 202121401081@stu.hebut.edu.cn (H.Z.); 202131404056@stu.hebut.edu.cn (K.L.)
[2] Zhejiang Huayun Electric Power Engineering Design & Consultation Co., Ltd., Hangzhou 310014, China; wfg8848@gmail.com
* Correspondence: hj2018@hebut.edu.cn; Tel.: +86-159-9636-5927

**Abstract:** The DC solid state transformer (DCSST) is a crucial component for connecting buses of different voltage levels in the DC distribution grid. This paper proposes a Si IGBT/SiC MOSFET hybrid isolated bidirectional DC–DC converter and an optimized modulation strategy (OMS) to reduce the losses and costs of DCSST. Based on the analysis of topology and operating principles, a duty-cycle modulation strategy is proposed and the converter is modeled by the time domain analysis (TDA) method. Through the analysis of switching characteristics, an optimization problem is established, which aims to reduce the conduction losses of switches while ensuring zero-voltage switching (ZVS) for all switches and low-current turn-off for IGBTs simultaneously. The optimization problem is solved by the augmented Lagrangian genetic algorithm (ALGA), and the OMS for the proposed converter is deduced. Finally, a 2 kW experimental prototype with the primary voltage of 405–495 V and the secondary voltage of 150 V is built to verify the effectiveness of the proposed topology and OMS. The switching costs of the proposed converter is reduced by 27.3% and the efficiency is improved by up to 4.04% compared to the existing method.

**Keywords:** hybrid-switch DC–DC converter; duty-cycle modulation; optimized modulation strategy; switching characteristics; DC solid state transformer





## 1. Introduction

In recent years, the DC distribution grid has received widespread attention due to the ease of access to energy storage and renewable energy systems, cost reduction, and improvement of power conversion efficiency [1,2]. A typical structure of a DC distribution grid shown in Figure 1 contains both a low-voltage DC (LVDC) bus and a medium-voltage DC (MVDC) bus. The DC solid state transformer (DCSST) is a crucial component for connecting the two buses [2,3]. The dual active bridge (DAB) DC–DC converter, with the advantages of bidirectional flow capability, easy soft switching, and high modularity, is the core circuit of DCSST [4]. The circuit structure of the DAB consists of the primary and secondary H-bridges, a high-frequency transformer, and a series inductor. Limited by the withstand voltage level of the single semiconductor switch, the topology of DCSST mostly adopts the DAB with the structure of input series output parallel (referred to as ISOP-DAB) in practice [5].

The basic modulation strategy for DAB is single phase-shift (SPS) modulation, where the magnitude and flow of power are controlled by adjusting the external phase-shift angle between the H-bridges. The SPS modulation strategy is easy to realize zero-voltage switching (ZVS) with middle and high power, and it especially can realize ZVS over the full power range under the unit voltage conversion ratio. However, under the non-unit voltage conversion ratio, the conduction losses, current stress, and reactive power of the DAB increase

and the switches lose ZVS in the low-power region, which deteriorates the efficiency of the converter. To address the above problems, extensive research has been carried out on modulation and optimized strategies. By increasing the internal phase-shift angles within the primary or secondary H-bridges of DAB, the modulation strategies of extended phase shift (EPS) [6,7], dual phase shift (DPS) [8,9], and triple phase shift (TPS) [10–14] have been successively proposed, which reduce the conduction losses of the converter and expand the ZVS range with the suitable combination of phase-shift angles. On the other hand, Refs. [15–17] proposed the asymmetric duty modulation (ADM) strategy by regulating the duty cycle of driving pulses of switches, which can achieve similar operating characteristics to phase-shift modulation. Further, Refs. [18,19] proposed the duty cycle plus phase-shift modulation by combining phase-shift modulation and ADM, which is expected to achieve even better operating characteristics since the degree of freedom of the modulation is increased to five.

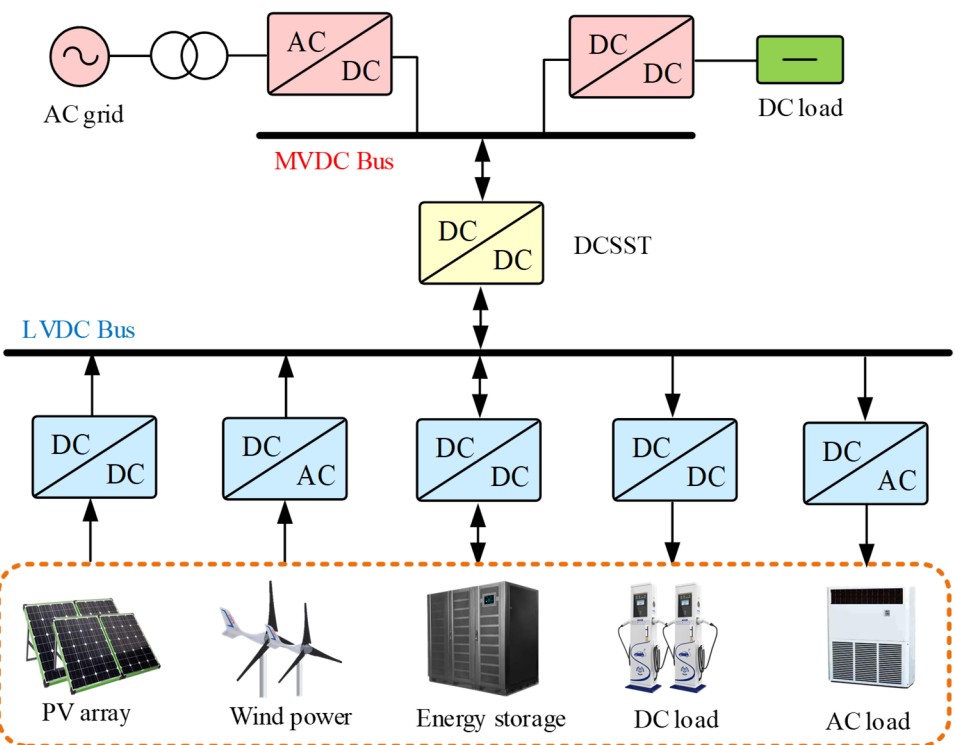

**Figure 1.** Typical structure diagram of DC distribution grid.

The optimization of the converter can be regarded as a problem of constrained non-linear optimization, and the effect of optimization depends on the accuracy of the model of converter, the constraints, and the optimized algorithm [20]. The modeling methods for DAB converter are usually divided into two types: time domain analysis (TDA) method [6–8,12,14,16,17] and harmonic component analysis (HCA) method [9,15,18,19]. The TDA method can deduce detailed expressions for parameters such as voltage and current based on dividing the operating regions and the operating modes of each region. The modeling accuracy of the TDA method is high, but the division of the operating regions and operating modes under complex modulation strategy is complicated and the method has difficulty achieving unified modeling. The expressions for the switching functions and voltages under the HCA method are expressed in the form of Fourier series, avoiding the need to analyze complex operating modes. However, the modeling accuracy of the HCA method depends on the maximum order of the Fourier series, and the model is usually not accurate enough under non-unit voltage conversion ratio or asymmetric modulation. The optimized objective of DAB is usually chosen as current stress [8,14,15], rms value of current [7,12,13,18,19], or reactive power [9], and the constraints are usually chosen as

ZVS constraints. The choice of optimized algorithm depends on the complexity of the model. When the mathematical expression of the model is simple, the analytical solution of the model is usually obtained by algorithms such as the Lagrange multiplier method [8]. When the model is more complex, convex optimization algorithms or heuristic algorithms such as genetic algorithm (GA), particle swarm optimization (PSO), and Q-learning algorithm [10,11] are usually used to solve the numerical solution of the model.

By comparing existing optimized modulation strategies, it is found that the optimized modulation strategies are mostly optimized for the operating characteristics such as current stress, rms value of current, and switching characteristics in the low-power and medium-power regions, and still use SPS modulation in the high-power region [12–15,18,19]. Therefore, the turn-off current is high in the high-power region. For the Si IGBTs, it generates large trailing current at high turn-off current, which causes high turn-off losses and deterioration of efficiency [20]. Furthermore, the sizes of passive components such as inductors and transformers are reduced by increasing the switching frequency. The switching frequency usually cannot be too high in order to ensure that the switches are reliably turned off at high turn-off current, which limits the size for components and power density. However, little attention has been paid to the turn-off current on the optimized modulation strategies in the current research. As a result, SiC MOSFETs are mostly used to reduce deterioration of efficiency in the high-power region. However, the problem of efficiency degradation still exists [12–15,18,19] and increases the costs of the converter. Utilizing Si IGBT/SiC MOSFET hybrid circuit structure and making Si IGBTs realize low-current turn-off by means of suitable modulation strategy is an effective way to solve the above problems, while there is a lack of related research.

In order to reduce the losses and costs of DCSST, this paper proposes a Si IGBT/SiC MOSFET hybrid isolated bidirectional DC–DC converter (referred to as hybrid-switch DC–DC converter or HSDC) and an optimized modulation strategy (OMS). The rest of this paper is organized as follows. In Section 2, the topology and operating principles of HSDC are introduced. In Section 3, the duty-cycle modulation strategy for HSDC is proposed and the switching characteristics are analyzed. In Section 4, the optimized modulation strategy (OMS) for HSDC is presented. In Section 5, a prototype is established to validate the proposed topology and optimized modulation strategy. Finally, conclusions are drawn in Section 6.

## 2. Hybrid-Switch DC–DC Converter with Three-Phase Medium-Frequency Transformer

### 2.1. Topology

Figure 2a shows the topology of the proposed hybrid-switch DC–DC converter. The primary side consists of three series-connected H-bridges and the secondary side is a three-phase half bridge. A three-phase medium-frequency transformer with delta connection on the secondary side is located between the primary and secondary bridges. $n$ is the transformer ratio. $V_{11}$, $V_{12}$, and $V_{13}$ are the DC voltages of the primary bridges. $V_1$ and $V_2$ are the DC port voltages. $I_1$ and $I_2$ are the DC port currents. $C_{11}$, $C_{12}$, $C_{13}$, and $C_2$ are the DC capacitors. $L_1$, $L_2$, and $L_3$ are the equivalent series inductances (sum of leakage inductances of transformer and series inductances). $u_{p1}$, $u_{p2}$, and $u_{p3}$ are the AC port voltages of the primary bridges. $u_{s1}$, $u_{s2}$, and $u_{s3}$ are the AC port voltages of the secondary bridge. $i_{p1}$, $i_{p2}$, and $i_{p3}$ are the AC port currents of the primary bridges. $i_{s1}$, $i_{s2}$, and $i_{s3}$ are the linear currents of the secondary bridge. $S_{pij}$ ($i = 1, 2, 3; j = 1, 2, 3, 4$) are the switches of the primary bridges, where $S_{pij}$ ($i = 1, 2, 3; j = 1, 3$) are defined as the upper switches with SiC MOSFETs, and $S_{pij}$ ($i = 1, 2, 3; j = 2, 4$) are defined as the lower switches with Si IGBTs. $S_{sk}$ ($k = 1, 2, …, 6$) are the switches with SiC MOSFETs of the secondary bridge.

In applications with higher voltage or current, the HSDC can be regarded as the submodule of the DCSST. Due to the voltage withstand level of the switches, the submodules usually need connecting in series or parallel. The number of submodules depends on the voltage, current, and transmitted power.

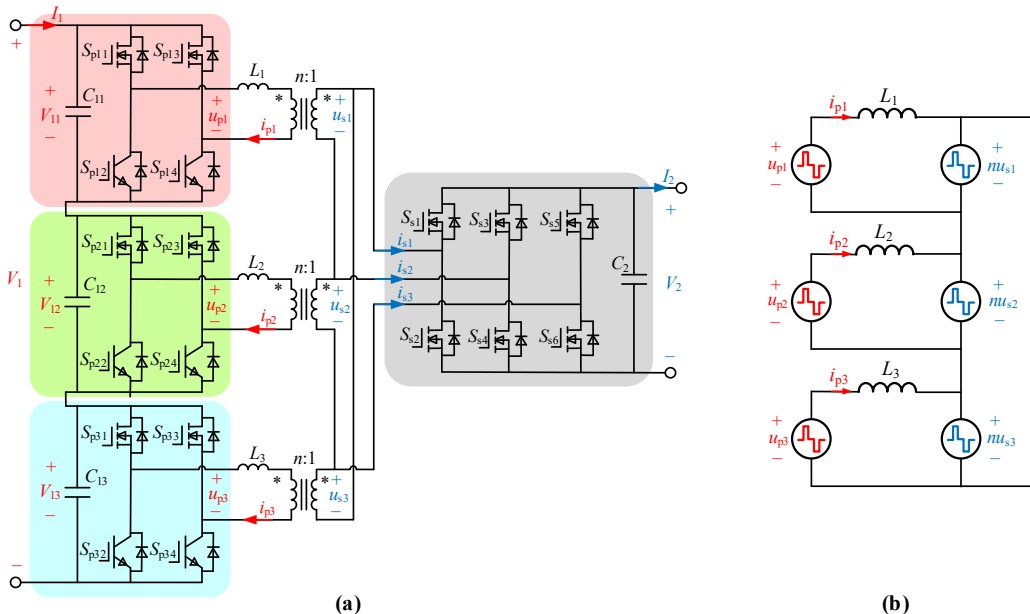

**Figure 2.** Topology and equivalent circuit of hybrid-switch DC–DC converter. (**a**) Topology; (**b**) equivalent circuit (The direction of the arrows indicates the positive direction of the currents, and "*" indicates the homonymous end of the transformer).

## 2.2. Operating Principles

Figure 2b shows the equivalent circuit diagram of HSDC. The primary bridges are equivalent to three AC voltage sources. The secondary bridge is equivalent to a delta connected three-phase AC voltage source, and the three-phase voltages are equivalent to the primary side as $nu_{s1}$, $nu_{s2}$, and $nu_{s3}$. Since the magnetizing inductances of the transformer are usually much larger than the equivalent series inductances, the magnetizing inductance can be neglected and the transformer is represented by the equivalent series inductance. According to the equivalent circuit, the relationship between voltages and currents is deduced as

$$
\begin{cases}
L_1 \frac{di_{p1}}{dt} = u_{p1} - nu_{s1} \\
L_2 \frac{di_{p2}}{dt} = u_{p2} - nu_{s2} \quad , \\
L_3 \frac{di_{p3}}{dt} = u_{p3} - nu_{s3}
\end{cases}
\tag{1}
$$

where the AC port voltages can be expressed by switching function. The switching function is defined as (2), and the AC port voltages are expressed as (3) and (4), respectively.

$$
s_{pij}(t) \text{ or } s_{sk}(t) =
\begin{cases}
1, & \text{when } S_{pij} \text{ or } S_{sk} \text{ conducts} \\
0, & \text{when } S_{pij} \text{ or } S_{sk} \text{ blocks}
\end{cases}
\tag{2}
$$

$$
u_{pi} = \frac{V_1\left(s_{pi1}(t) - s_{pi3}(t)\right)}{3}, i = 1, 2, 3
\tag{3}
$$

$$
\begin{cases}
u_{s1} = V_2(s_{s1}(t) - s_{s3}(t)) \\
u_{s2} = V_2(s_{s3}(t) - s_{s5}(t)) \\
u_{s3} = V_2(s_{s5}(t) - s_{s1}(t))
\end{cases}
\tag{4}
$$

## 3. Duty-Cycle Modulation for Hybrid-Switch DC–DC Converter

To simplify the analysis, the following assumptions are made: (1) All the switches, inductors, capacitors, and transformer are ideal. (2) The voltages across $C_{11}$, $C_{12}$, and $C_{13}$ are

balanced at $V_1/3$. (3) $L_1$, $L_2$, and $L_3$ are the same and equal to $L$. The waveforms of driving pulses, voltages, and current under duty-cycle modulation (DCM) are shown in Figure 3, where $T_s$ is the switching period. The driving pulses of the upper and lower switches of each half-bridge are complementary. The duty cycle of driving pulses for the primary upper switches is $D$, and the duty cycle of driving pulses for the secondary switches is constant at 1/2. The phase-shift between the two half-bridges of each H-bridge is $T_s/2$ and the phase-shift between H-bridges of adjacent phases is $T_s/3$. The phase-shift between adjacent phases of the three-phase half bridge is $T_s/3$ as well. The phase-shift between the primary and secondary switches is variable. Taking A-phase as an example, the phase-shift between $S_{p11}$ and $S_{s1}$ is $D_f T_s$, with $D_f$ defined as the phase-shift ratio. The range of values for the two control parameters is $0 \leq D \leq 1/2$, $0 \leq D_f \leq 1$. The converter operates in forward mode for $0 \leq D_f \leq \frac{1}{2}$, while the converter operates in reverse mode for $1/2 \leq D_f \leq 1$. In particular, the modulation is SPS modulation when $D$ is constant at 1/2.

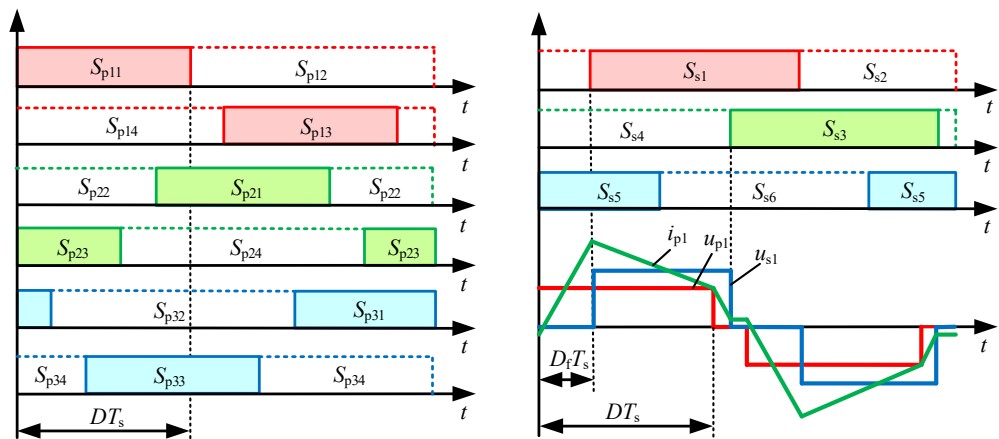

**Figure 3.** Waveforms of driving pulses, voltages, and current of HSDC with DCM.

### 3.1. Modeling

The TDA method is used for modeling in this paper. Without loss of generality, the analysis of HSDC is carried out with A-phase as an example, since the parameters of three phases are consistent. When the converter operates in the forward mode, the range of values for the control parameters can be divided into five operating regions, as shown in Figure 4, and the ranges of $D$ and $D_f$ for each region are shown in Table 1. The operating waveforms for each operating region are shown in Figure 5.

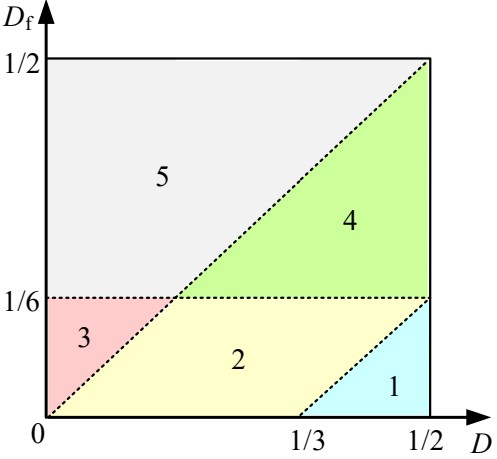

**Figure 4.** Division of operating regions in the forward mode (The numbers indicate the numbers of operating regions).

**Table 1.** Ranges of $D$ and $D_f$ for operating regions.

| Region | Range of $D$ and $D_f$ |
| --- | --- |
| 1 | $1/3 \leq D \leq 1/2, 0 \leq D_f \leq D - 1/3$ |
| 2 | $0 \leq D \leq 1/2, \max\{0, D - 1/3\} \leq D_f \leq \min\{D, 1/6\}$ |
| 3 | $0 \leq D \leq 1/6, D \leq D_f \leq 1/6$ |
| 4 | $1/6 \leq D \leq 1/2, 1/6 \leq D_f \leq D$ |
| 5 | $0 \leq D \leq 1/2, \max\{1/6, D\} \leq D_f \leq 1/2$ |

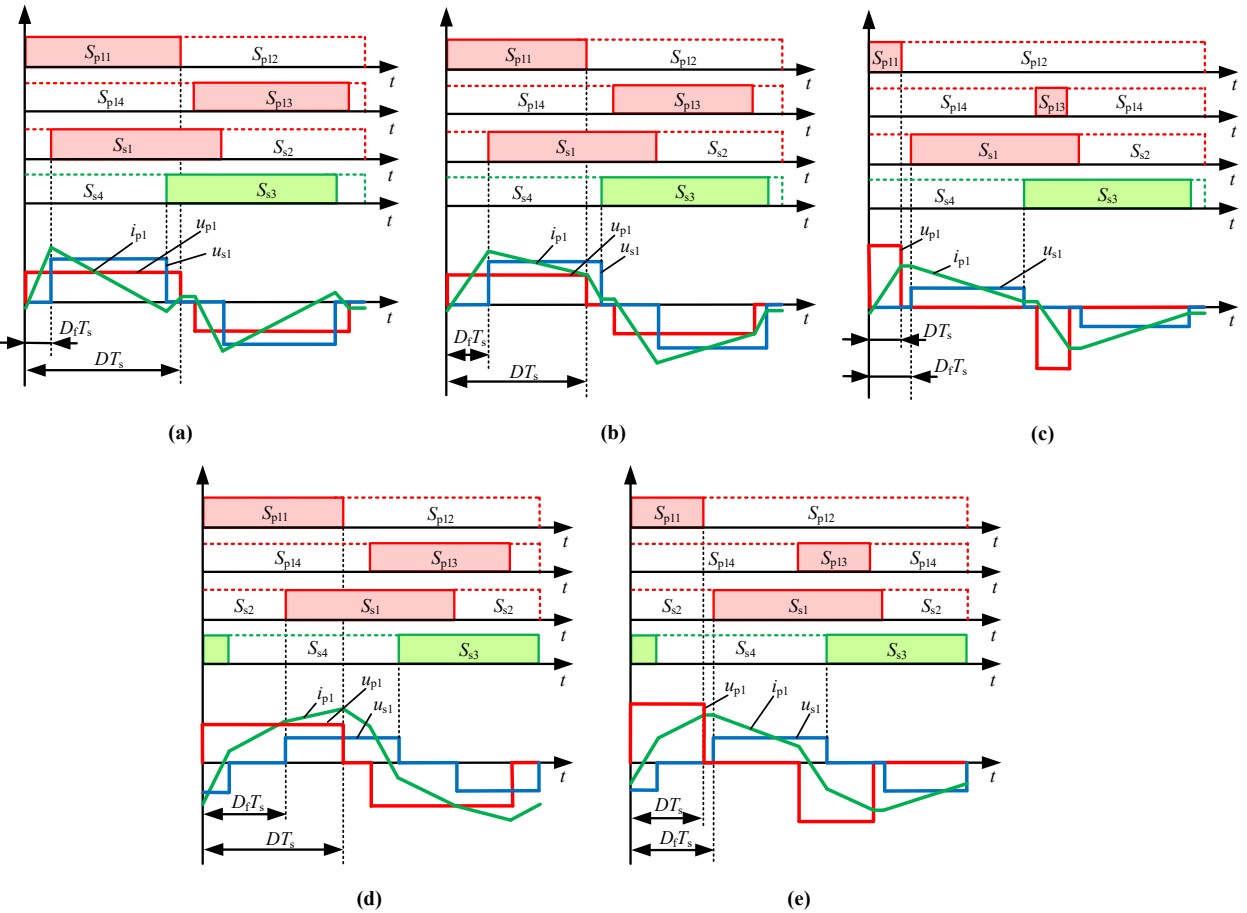

**Figure 5.** Waveforms of driving pulses, voltages, and current of operating regions in the forward mode: (**a**) Region 1; (**b**) Region 2; (**c**) Region 3; (**d**) Region 4; (**e**) Region 5.

Region 1 is used as an example for illustration. According to the operating waveforms shown in Figure 5a, the switching cycle of Region 1 can be divided into eight operating modes, and the analysis can be simplified for four operating modes due to the symmetry of the waveforms. According to (1), the expression of $i_{p1}(t)$ for half switching cycle is shown in Table 2. From the symmetry of the waveform, the relation is obtained as (5). Then, the expression of $i_{p1}$ at each time point can be calculated as shown in Table 3, where the voltage conversion ratio $M$ is defined as $3\, nV_2/V_1$.

$$i_{p1}(0) + i_{p1}\left(\frac{T_s}{2}\right) = 0 \tag{5}$$

**Table 2.** Expression of $i_{p1}(t)$ for each operating mode of Region 1.

| Mode | Range of Time | $u_{p1}$ | $u_{p2}$ | Expression of $i_{p1}(t)$ |
|------|---------------|----------|----------|----------------------------|
| 1 | $[0, D_f T_s]$ | $\frac{V_1}{3}$ | $0$ | $i_{p1}(0) + \frac{V_1}{3L}t$ |
| 2 | $\left[D_f T_s, \left(D_f + \frac{1}{3}\right)T_s\right]$ | $\frac{V_1}{3}$ | $V_2$ | $i_{p1}(D_f T_s) + \frac{V_1 - 3nV_2}{3L}(t - D_f T_s)$ |
| 3 | $\left[\left(D_f + \frac{1}{3}\right)T_s,\ DT_s\right]$ | $\frac{V_1}{3}$ | $0$ | $i_{p1}\left[\left(D_f + \frac{1}{3}\right)T_s\right] + \frac{V_1}{3L}\left[t - \left(D_f + \frac{1}{3}\right)T_s\right]$ |
| 4 | $\left[DT_s, \frac{T_s}{2}\right]$ | $0$ | $0$ | $i_{p1}(DT_s)$ |

**Table 3.** Expression of $i_{p1}$ at each time point of Region 1.

| Time Point | Expression of $i_{p1}$ |
|------------|------------------------|
| $0$ | $-\frac{nV_2}{6f_sL}\left(\frac{3D}{M} - 1\right)$ |
| $D_f T_s$ | $-\frac{nV_2}{6f_sL}\left(\frac{3D}{M} - \frac{6D_f}{M} - 1\right)$ |
| $\left(D_f + \frac{1}{3}\right)T_s$ | $-\frac{nV_2}{6f_sL}\left(\frac{3D}{M} - \frac{6D_f}{M} - \frac{2}{M} + 1\right)$ |
| $DT_s$ | $-\frac{nV_2}{6f_sL}\left(-\frac{3D}{M} + 1\right)$ |
| $\frac{T_s}{2}$ | $-\frac{nV_2}{6f_sL}\left(-\frac{3D}{M} + 1\right)$ |

Based on the above analysis, the expression of the transmission power $P$ is calculated as

$$P = \frac{6}{T_s}\int_0^{\frac{T_s}{2}} u_{p1}(t)i_{p1}(t)\mathrm{d}t = \frac{nV_1V_2}{9f_sL}(6D_f - 3D + 1). \tag{6}$$

The expression of the rms value of the primary AC current $I_{rms}$ is calculated as

$$
\begin{aligned}
I_{rms} &= \sqrt{\frac{2}{T_s}\int_0^{\frac{T_s}{2}} \left[i_{p1}(t)\right]^2 \mathrm{d}t} \\
&= \frac{nV_2}{18Mf_sL}\sqrt{\frac{-108D^3 + 108MD^2 + 81D^2 - 216MDD_f}{-90MD + 216MD_f^2 + 72MD_f + 5M^2 + 8M}}
\end{aligned}. \tag{7}
$$

In order to simplify the expression for subsequent work, the expression is normalized. Define the power reference $P_B$ and current reference $I_B$ as

$$
\begin{cases}
P_B = P\big|_{D=\frac{1}{2}, D_f=\frac{1}{6}} = \frac{nV_1V_2}{18f_sL} \\
I_B = \frac{P_B}{V_1} = \frac{nV_2}{18f_sL}
\end{cases}. \tag{8}
$$

The transmission power and rms value of current for Region 1 are expressed by

$$
\begin{cases}
P = 2(6D_f - 3D + 1) \\
I_{rms} = \frac{1}{M}\sqrt{\dfrac{-108D^3 + 108MD^2 + 81D^2 - 216MDD_f}{-90MD + 216MD_f^2 + 72MD_f + 5M^2 + 8M}}
\end{cases}. \tag{9}
$$

Other operating regions can be modeled following the similar procedure. The power reference $P_B$ and current reference $I_B$ is the same as Region 1, and the normalized expressions of the transmission power and rms current are shown in Table 4.

**Table 4.** Normalized expressions of the transmission power and rms current.

| Region | Transmission Power | Rms Current |
|---|---|---|
| 1 | $2(6D_f - 3D + 1)$ | $\frac{1}{M}\sqrt{\begin{array}{l}-108D^3 + 108MD^2 + 81D^2 - 216MDD_f - 90MD + 216MD_f^2\\ +72MD_f + 5M^2 + 8M\end{array}}$ |
| 2 | $6(-3D^2 - 3D_f^2 + 6DD_f + D)$ | $\frac{1}{3M}\sqrt{\begin{array}{l}-6048M^2D^3 + 5832MD^3 - 912D^3 + 18144M^2D^2D_f - 13608MD^2D_f\\ +4752M^2D^2 - 3564MD^2 + 729D^2 - 18144MDD_f^2 - 9504M^2DD_f\\ +4536MDD_f - 1152M^2D + 270MD + 6048M^2D_f^3 - 1944MD_f^3\\ +4752M^2D_f^2 + 1150M^2D_f + 125M^2\end{array}}$ |
| 3 | $6D$ | $\frac{1}{M}\sqrt{-108MD^2 + 81D^2 + 5M^2 - 108D^3 - 18MD + 216MDD_f}$ |
| 4 | $-18D^2 - 36D_f^2 + 36DD_f + 6D + 6D_f - \frac{1}{2}$ | $\frac{1}{M}\sqrt{\begin{array}{l}216MD^3 - 108D^3 - 648M^2D_f - 108MD^2 + 81D^2 + 648MDD_f^2\\ +216MDD_f - 27MD + 9D - 432MD_f^3 + 108MD_f^2 - 18D_f + 9M^2\\ -5M + 2\end{array}}$ |
| 5 | $-18D_f^2 + 6D_f + 6D - \frac{1}{2}$ | $\frac{1}{M}\sqrt{\begin{array}{l}-108D^3 - 108MD^2 + 81D^2 + 216MDD_f - 18MD - 216MD_f^3\\ +108MD_f^2 - 18MD_f + 5M^2 + M\end{array}}$ |

Further, the reverse mode can be modeled following a similar procedure.

### 3.2. Switching Characteristics

#### 3.2.1. Turn-On Characteristics

If the current flows through the antiparallel diode the moment the switch is turned on, the switch is considered to realize ZVS, and the turn-on loss can be ignored at this time. Taking Region 1 as an example, the ZVS condition for the primary upper and lower switches is

$$i_{p1}(0) = 3\left(1 - \frac{3D}{M}\right) \leq 0. \tag{10}$$

The ZVS condition for the secondary switches is

$$i_{s1}(D_fT_s) = n[i_{p1}(D_fT_s) - i_{p3}(T_s)] = 6n\left(1 - \frac{1}{M}\right) \geq 0. \tag{11}$$

ZVS conditions for other operating regions can be deduced following a similar approach. Since the ZVS conditions are related to the voltage conversion ratio $M$, the turn-on characteristics differ with different $M$. The turn-on characteristics over the full power range with different voltage conversion ratios are shown in Figure 6, where the number on the contour lines denotes normalized power, and the blue-filled area indicates that all switches of the converter can realize ZVS (abbreviated as full-switch ZVS).

The converter can realize full-switch ZVS in Regions 1, 2, and 4. Comparing Figure 6a–c, the ZVS ranges of Regions 1, 2, and 4 gradually decrease with the increase of M and the full-switch ZVS can achieve over the full power range when $M \geq 1$. Comparing Figure 6a,d,e, Region 1 loses full-switch ZVS and the converter is not able to realize all-switch ZVS in the low-power region when $M < 1$.

#### 3.2.2. Turn-Off Characteristics

Taking the primary switches of A-phase as an example, the relationship between the driving pulse of $S_{p11}$ and $i_{p1}$ can reflect the relationship between the upper switch and the current flowing through it, while the relationship between the driving pulse of $S_{p14}$ and $i_{p1}$ can reflect the relationship between the lower switch and the current flowing through it. Region 2 is used as an example for illustration. As shown in Figure 7, $I_{p\_on1}$ and $I_{poff\_1}$ are the turn-on current and turn-off current of the upper switch, respectively, and $I_{p\_on2}$ and $I_{poff\_2}$ are the turn-on current and turn-off current of the lower switch, respectively. The turn-off current of the upper switch is equal in magnitude and opposite in direction to that of the lower switch, and vice versa. Therefore, the premise of low-current turn-off is that the switch realizes ZVS.

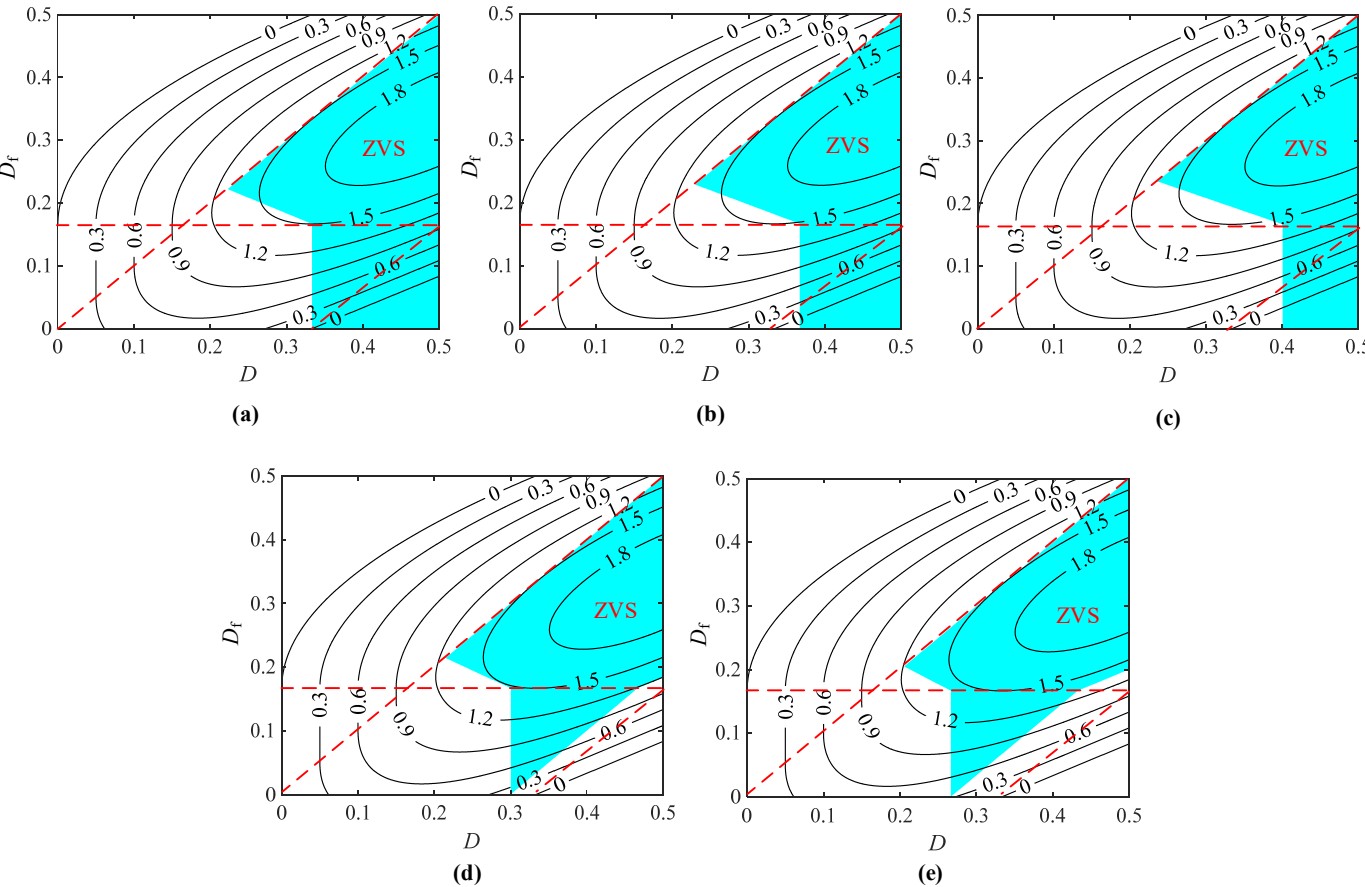

**Figure 6.** Turn-on characteristics over the full power range: (**a**) $M = 1$; (**b**) $M = 1.1$; (**c**) $M = 1.2$; (**d**) $M = 0.9$; (**e**) $M = 0.8$.

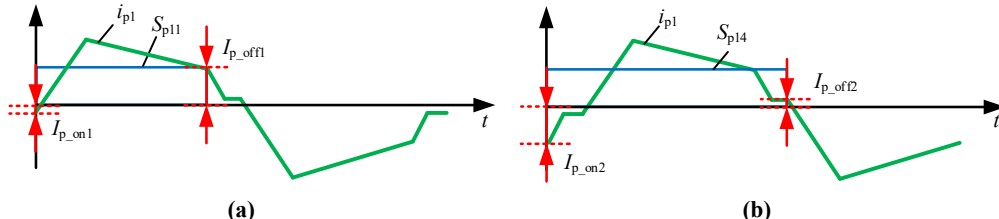

**Figure 7.** Diagram of switching characteristics in Region 2: (**a**) Upper switch; (**b**) Lower switch.

## 4. Optimized Modulation Strategy for Reducing Switching Losses

### 4.1. Optimized Modulation Strategy

#### 4.1.1. Optimization Problem

The optimization of the converter can be regarded as a problem of constrained nonlinear optimization. The rms value of the primary AC current is positively related to the conduction losses of switches and the winding losses of transformers, and minimizing the rms current can ensure the minimization of the conduction losses and winding losses. Considering the form of the expression, the optimized objective is set to minimize the square of the rms value of the current. Denote the rms of the current as $I_{\mathrm{rms}}(D, D_{\mathrm{f}})$, and the objective function is

$$\text{Min } f = I_{\mathrm{rms}}^2(D, D_{\mathrm{f}}). \tag{12}$$

The equation constraint for the optimization problem is the constraint of the transmission power. Denote the transmission power as $P(D, D_f)$, and the equation constraint is shown as (13), where $P_t$ is the target value of the transmission power.

$$P(D, D_f) = P_t \tag{13}$$

In practice, the parasitic capacitance of the switch needs to be considered to ensure that the switch realizes ZVS. If the energy stored in the inductor is greater than the minimum turn-on current the moment the switch is turned on, the switch can realize ZVS in practice [21,22]. The minimum turn-on current can be determined by

$$\begin{cases} \frac{1}{2}LI^2_{\text{p\_onmin}} = \frac{1}{2}(C_{\text{oss11}} + C_{\text{oss12}})\left(\frac{V_1}{3}\right)^2 \\ \frac{1}{2}n^2L\left(\frac{I_{\text{s\_onmin}}}{\sqrt{3}}\right)^2 = C_{\text{oss2}}V_2^2 \end{cases}, \tag{14}$$

where $I_{\text{p\_onmin}}$ and $I_{\text{s\_onmin}}$ are the minimum turn-on currents of the primary and secondary switches, respectively; $C_{\text{oss11}}$ and $C_{\text{oss12}}$ are the parasitic capacitances of the primary upper and lower switches, respectively; and $C_{\text{oss2}}$ is the parasitic capacitance of the secondary switches. Normalize the minimum turn-on current, and the ZVS constraints for the optimization problem are obtained, as shown as (15), where $I_{\text{p\_on1}}$ and $I_{\text{p\_on2}}$ are the turn-on currents of the primary upper and lower switches, respectively, and $I_{\text{s\_on}}$ is the turn-on current of the secondary switch.

$$\begin{cases} I_{\text{p\_on1}} \leq -\frac{18f_s\sqrt{L(C_{\text{oss11}}+C_{\text{oss12}})}}{M} \\ I_{\text{p\_on2}} \leq -\frac{18f_s\sqrt{L(C_{\text{oss11}}+C_{\text{oss12}})}}{M} \\ I_{\text{s\_on}} \leq -18f_s\sqrt{6LC_{\text{oss2}}} \end{cases} \tag{15}$$

The primary lower switches are Si IGBTs, which need to achieve low-current turn-off to ensure reliable switching and reduce turn-off losses. The turn-off current is limited by setting a suitable maximum turn-off current, as shown in (16), and the constraint of turn-off current is shown as (16), where $I_{\text{p\_offmax}}$ is the maximum turn-off current of the primary lower switches.

$$I_{\text{p\_off2}} \leq \frac{I_{\text{p\_offmax}}}{I_B} \tag{16}$$

In addition, the range of values for $D$ and $D_f$ are also constraints, which are determined by the operating regions. The optimized model obtained by the above process is complex, and it is difficult to find the analytical solution of the model. As a result, the augmented Lagrangian genetic algorithm (ALGA) is chosen to solve the numerical solution in this paper. Define $X = [D, D_f]$, and the proposed optimized model can be organized into the standard form, shown as

$$\text{Min } f(X)$$
$$s.t. \begin{cases} c_i(X) \leq 0, i = 1, 2, \ldots, m \\ ceq(X) = 0 \end{cases}, \tag{17}$$

where $c_i(X)$ and $ceq(X)$ denote the inequality and equation constraints, respectively; $m$ is the number of inequality constraints. The expression for the fitness function of ALGA is obtained as shown in (18), where $\lambda$ is the Lagrange multiplier vector whose component $\lambda_i$ ($i = 1, 2, \ldots, m + 1$) is non-negative; $s$ is the shift vector whose component $s_i$ ($i = 1, 2, \ldots, m$) is also non-negative; $\rho$ is a positive penalty parameter.

$$\text{fitness}(\boldsymbol{X}, \boldsymbol{\lambda}, \boldsymbol{s}, \rho) = f(\boldsymbol{X}) - \sum_{i=1}^{m} \lambda_i s_i \log[s_i - c_i(\boldsymbol{X})] + \lambda_{m+1} ceq(\boldsymbol{X}) + \frac{\rho}{2}\left[ceq(\boldsymbol{X})\right]^2 \quad (18)$$

Define $\boldsymbol{C} = [n, M, P_t, fs, L, C_{oss11}, C_{oss12}, C_{oss2}, I_{p\_offmax}]$ as the parameter vector. The flow of model solving for ALGA based on the fitness function is shown in Figure 8, where $k$ is the number of iterations, $k_{max}$ is the maximum number of iterations, and $\varepsilon$ is the solving accuracy. The exact flow of the algorithm is as follows:

Step 1: Input $\boldsymbol{C}$, $\boldsymbol{\lambda}$, $\boldsymbol{s}$, $\rho$, $\varepsilon$, and $k$max, initialize $\boldsymbol{X_0}$, and calculate the fitness($\boldsymbol{X_0}$), where $\boldsymbol{X_0}$ is the iterative initial value of the ALGA. In practical calculation, the value of $\boldsymbol{X_0}$ is related to the number of iterations and is almost independent of the iteration result. For simplicity, the value of $\boldsymbol{X_0}$ in this paper is chosen to be a random value within the range of values.

Step 2: $\boldsymbol{X_{k-1}}$ is used as the input for the process of replication, crossover, and mutation, completing the kth iteration to obtain $\boldsymbol{X_k}$. Next, calculate the fitness ($\boldsymbol{X_k}$).

Step 3: If the absolute value of the difference between the fitness of the *k*th iteration and the $k-1$th iteration is less than $\varepsilon$ or $k$ reaches the maximum number of iterations, the iteration is stopped; otherwise, Step 2 and Step 3 are repeated.

Step 4: Test whether the iteration result $\boldsymbol{X_k}$ satisfies the constraints; if it does, then output $\boldsymbol{X} = \boldsymbol{X_k}$, otherwise the SPS modulation strategy is used, and $\boldsymbol{X} = [1/2, (P_t + 1)/12]$.

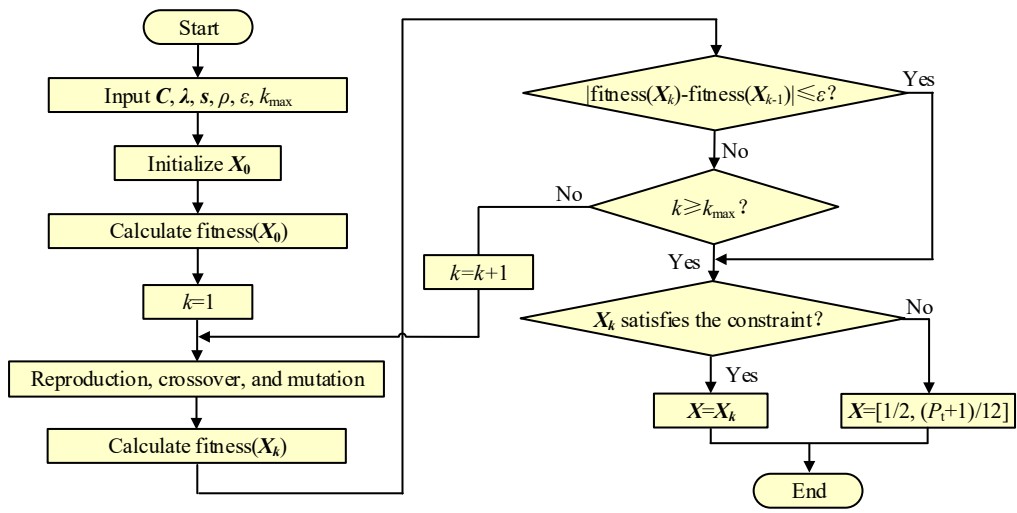

**Figure 8.** Flowchart of solving optimization problems with ALGA.

### 4.1.2. Optimized Modulation Strategy

According to the analysis in Section 3.2.1, the converter can realize full-switch ZVS over the full power range in Regions 1, 2, and 4 when $M \geq 1$, and realize full-switching ZVS for medium- and high-power in Regions 2 and 4 when $M < 1$. Therefore, a case-by-case discussion is needed for the value of $M$. The flow of the algorithm for OMS is shown in Figure 9, where $\boldsymbol{X_1}$, $\boldsymbol{X_2}$, and $\boldsymbol{X_4}$ are the optimized results for Regions 1, 2, and 4, respectively. The exact flow of the algorithm is as follows:

Step 1: Input $C$, $\boldsymbol{\lambda}$, $\boldsymbol{s}$, $\rho$, $\varepsilon$, and $k_{max}$.

Step 2: If $M \geq 1$, solve the optimization problems for Regions 1, 2, and 4; otherwise solve the optimization problems for Regions 2 and 4.

Step 3: Calculate the $I_{rms}$ of the optimized results, and take the optimization result whose $I_{rms}$ is smallest as the result of the OMS.

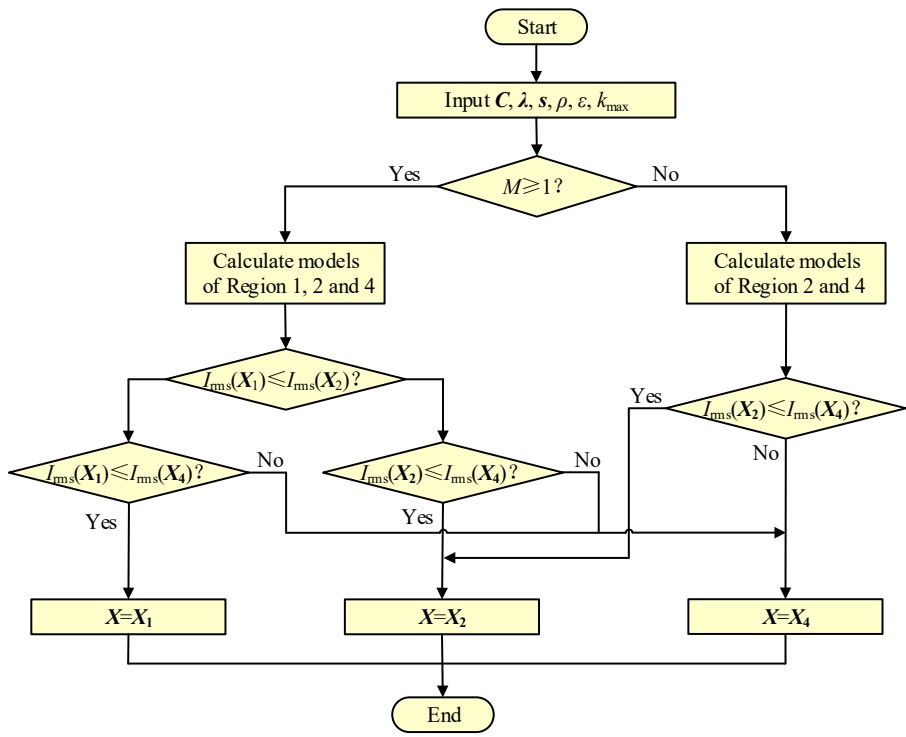

**Figure 9.** Flowchart of algorithms for OMS.

The MATLAB program is used to solve the above algorithm, where ALGA is mainly written based on the built-in function ga. The relevant parameters of HSDC are given in Table 5. Since the nanocrystalline core of the transformer typically operates at 10–100 kHz, the switching frequency of the converter is chosen to be 50 kHz. The transformer ratio is undetermined and needs to be designed according to the OMS. By considering the actual winding of the transformer, the ratio is accurate to one decimal place. Based on the above OMS, it is solved for transformer ratios of 0.9, 1.0, 1.1, 1.2, and 1.3, and the rms values of the currents for different transformer ratios are obtained, shown in Figure 10. When the primary voltage is different, the relationship of the rms values of the currents under different transformer ratios is different. However, the rms values of the currents are always at low level when $n$ is taken as 1.1. Therefore, the transformer ratio is determined to be 1.1. The optimized control parameters under the OMS are shown in Figure 11.

**Table 5.** Parameters of HSDC and ISOP-DAB.

| Parameters | Value | |
|---|---|---|
| | **HSDC** | **ISOP-DAB** |
| Primary DC voltage ($V_1$) | 450 V ($\pm$10%) | |
| Secondary DC voltage ($V_2$) | 150 V | |
| Power ($P$) | 1000–2000 W | |
| Switching frequency ($f_s$) | 50 kHz | |
| Transformer ratio ($n$) | 0.9–1.3 | 2 |
| Equivalent series inductance ($L$) | 30 μH | 70 μH |
| Si IGBT | IHW50N65R5 | |
| SiC MOSFET | IMW120R045M1 | |

To verify the effectiveness of OMS compared to existing modulation strategies, the switching characteristics of the HSDC under OMS are compared with those of ISOP-DAB under SPS modulation. To ensure the same number of switches, the topology of ISOP-DAB is shown in Figure 12, and the related parameters are shown in Table 5. ISOP-DAB utilizes SPS modulation, which can realize ZVS over the full power range under the unit voltage

conversion ratio. Therefore, the optimal transformer ratio for ISOP-DAB is 2, which is chosen according to the unit voltage conversion ratio.

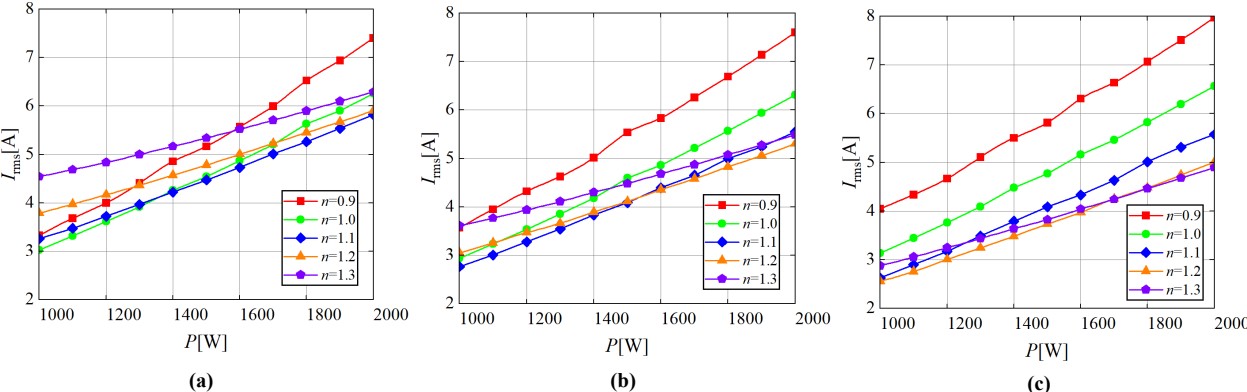

**Figure 10.** Rms values of the primary currents for different transformer ratios: (**a**) $V_1$ = 405 V; (**b**) $V_1$ = 450 V; (**c**) $V_1$ = 495 V.

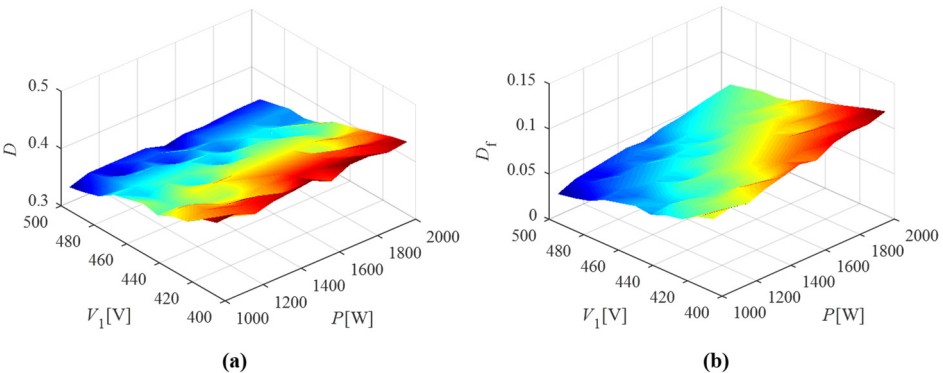

**Figure 11.** Optimal control parameters of OMS: (**a**) Duty cycle; (**b**) Phase-shift ratio.

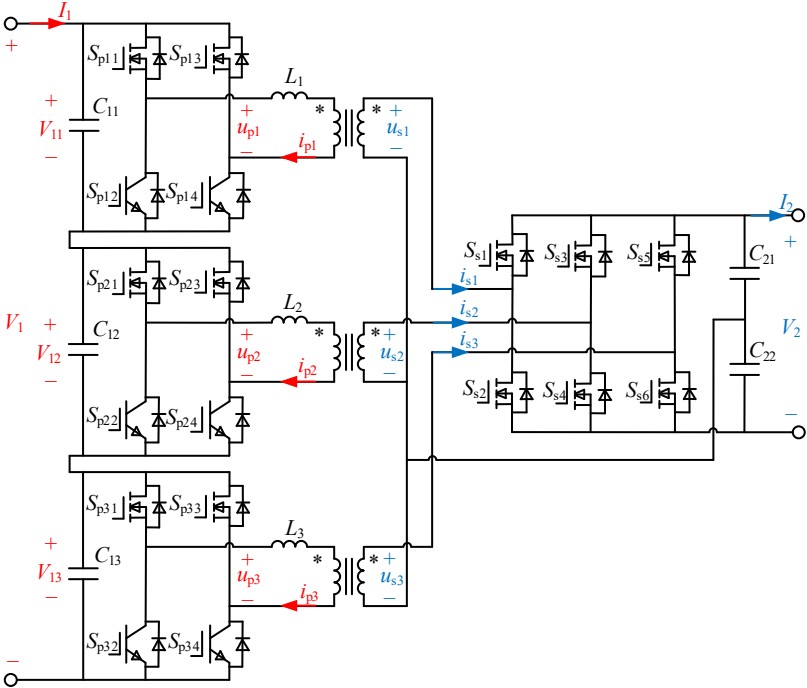

**Figure 12.** Topology of ISOP-DAB (The direction of the arrows indicates the positive direction of the currents, and "*" indicates the homonymous end of the transformer).

The comparison of the theoretical values of the turn-off currents for the primary switches is shown in Figure 13. For the upper switch, there is not much difference in the turn-off currents between the two converters when the primary voltage is 495 V. As the primary voltage drops, the turn-off current of HSDC will be lower than that of ISOP-DAB, and the difference between the two converters gradually increases. For the lower switch, the turn-off currents of HSDC are much lower than those of ISOP-DAB.

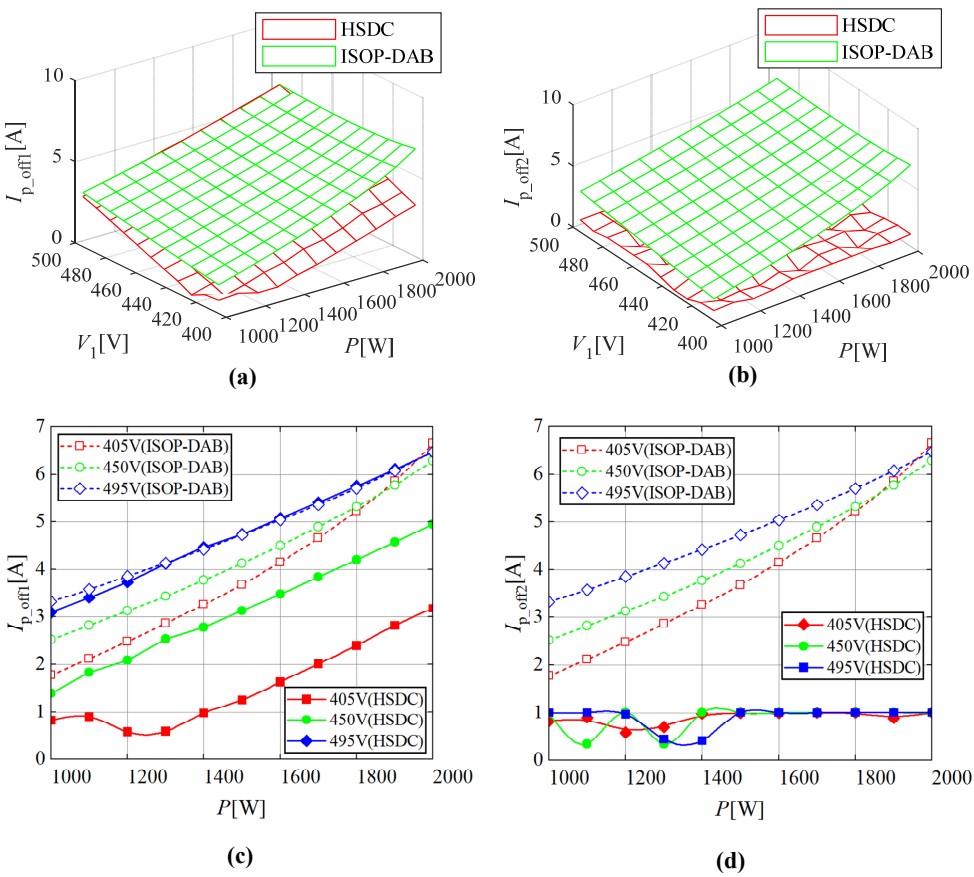

**Figure 13.** Theoretical turn-off current of primary switches of HSDC and ISOP-DAB. (**a**) Three-dimensional curves of upper switches; (**b**) Three-dimensional curves of lower switches; (**c**) Two-dimensional curves of upper switches; (**d**) Two-dimensional curves of lower switches.

### 4.2. Closed-Loop Control

When the converter operates in forward mode, a closed-loop control strategy is proposed, and the block diagram of closed-loop control strategy is shown in Figure 14. The optimization problem is complex to solve and the optimized result is numerical solutions. Therefore, the optimized duty cycle $D$ is calculated offline and stored in the memory of the microcontroller in advance, then $D$ is obtained online by the look-up table method. In the table of optimized duty cycle, the row data are varied by $V_1$ and the column data are varied by $P$. The steps of $V_1$ and $P$ are $\Delta V_1$ and $\Delta P$, respectively. Since the optimized results are discrete, the interpolation method is used to adjust $D$ [23]. The closed-loop control strategy is as follows:

Sample the secondary DC voltage $V_2$ and the secondary DC current $I_2$, and multiply the two to obtain the power $P$. Sample $V_{11}$, $V_{12}$, and $V_{13}$, and add the three to obtain the primary voltage $V_1$. $P$ and $V_1$ are looked up in the table to obtain the optimized duty cycle $D$. The shift ratio $D_f$ is obtained by closed-loop of voltage. Specifically, the reference value of the secondary voltage $V_{2ref}$ makes the difference with the actual value, and the difference is passed through the PI controller to obtain $D_f$. $D$ and $D_f$ are input into the pulse generator to obtain the driving pulse of each switch to complete the process of closed-loop control.

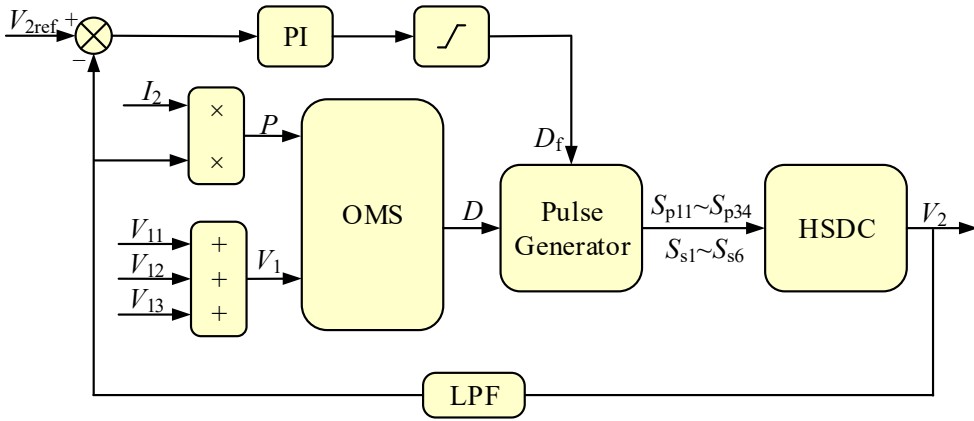

**Figure 14.** Block diagram of closed-loop control strategy.

## 5. Experimental Results

A prototype, as shown in Figure 15, is established to verify the proposed topology and OMS. The controller is TMS320F28379D of Texas Instruments. The material of the core for the transformer is nanocrystalline 1K107 and the number of turns on the primary and secondary sides of the transformer are 11 and 10, respectively. The parameters of the prototype are shown in Table 5.

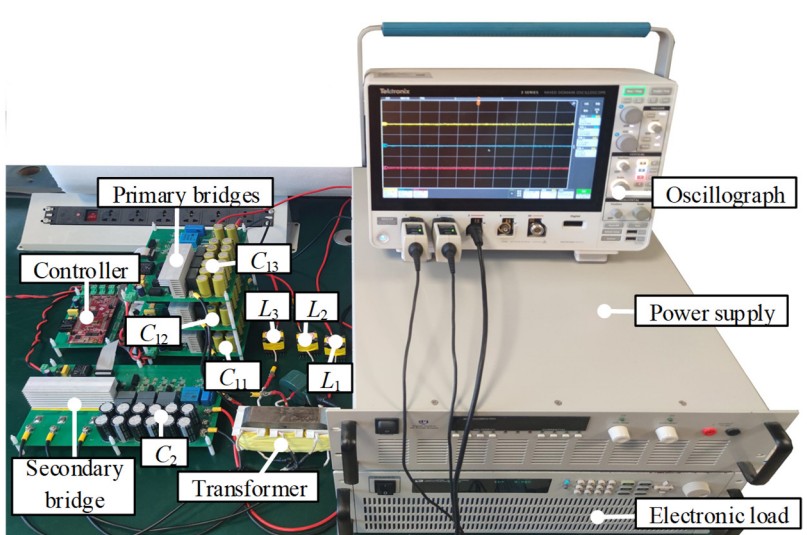

**Figure 15.** Experimental prototype of HSDC.

For the proposed HSDC, 1/3 of the switches are Si IGBTs, and the rest are SiC MOSFETs. In the case of the switches used in the experimental platform, there is a reduction of about 27.3% in the costs of switches compared to the current all-SiC MOSFET configuration.

The experimental waveforms of $u_{p1}$, $u_{s1}$, and $i_{p1}$ with the primary voltage of 405 V are shown in Figure 16. At the power of 1000 W, the converter operates in Region 1, when both the primary upper and lower switches can realize ZVS and low-current turn-off. As the power increases, the operating region of the converter transitions from Region 1 to Region 2. At the power of 2000 W, the converter operates in Region 2, when both the upper and lower switches can realize ZVS, and only the lower switch realizes low-current turn-off.

The switching characteristics of the primary upper and lower switches are analyzed in detail below, and the experimental waveforms of ISOP-DAB under SPS modulation are used as a comparison. The parameters of ISOP-DAB are shown in Table 4. The same nanocrystalline 1K107 is used as the material for the transformer, and the number of turns on the primary and secondary sides of the transformer are 20 and 10, respectively. The

switching characteristics of the primary upper switch are reflected by the voltage between the gate and the source $v_{gs}$, the voltage between the drain and the source $v_{ds}$, and the current flowing through the switch; the switching characteristics of the primary lower switch are reflected by the voltage between the gate and emitter $v_{ge}$, the voltage between the collector and emitter $v_{ce}$, and the current flowing through the switch. Since the switching characteristics of the primary upper and lower switches of the ISOP-DAB are basically the same under phase-shift modulation, only the waveforms of the upper switch are demonstrated. This paper selects $S_{p11}$ and $S_{p14}$ for illustration.

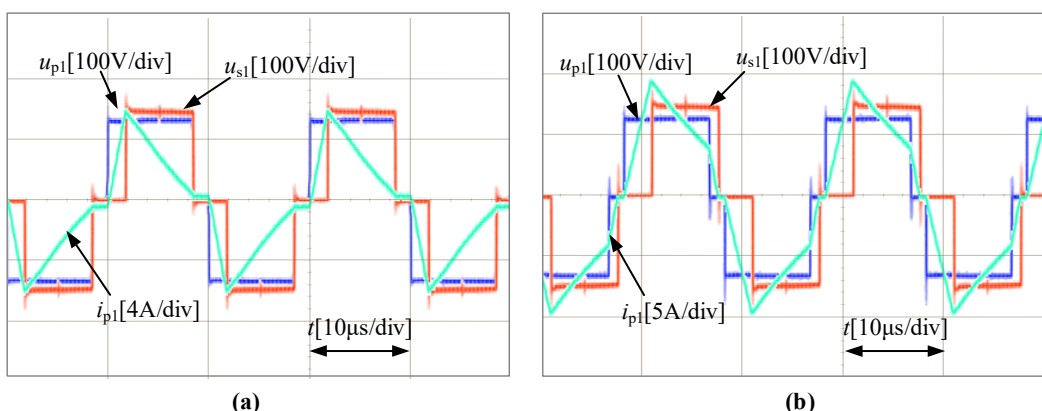

**Figure 16.** Operating waveforms for primary voltage 405 V of HSDC (**a**) 1000 W; (**b**) 2000 W.

The switching characteristics for the primary voltage of 405 V and power of 1000 W are given in Figure 17. The primary switches of both HSDC and ISOP-DAB realize ZVS. The turn-off current of both the upper and lower switches for HSDC is 0.4 A, while the turn-off current of both the upper and lower switches for the ISOP-DAB is 2.4 A. The turn-off losses of the primary switches of ISOP-DAB are higher than those of HSDC.

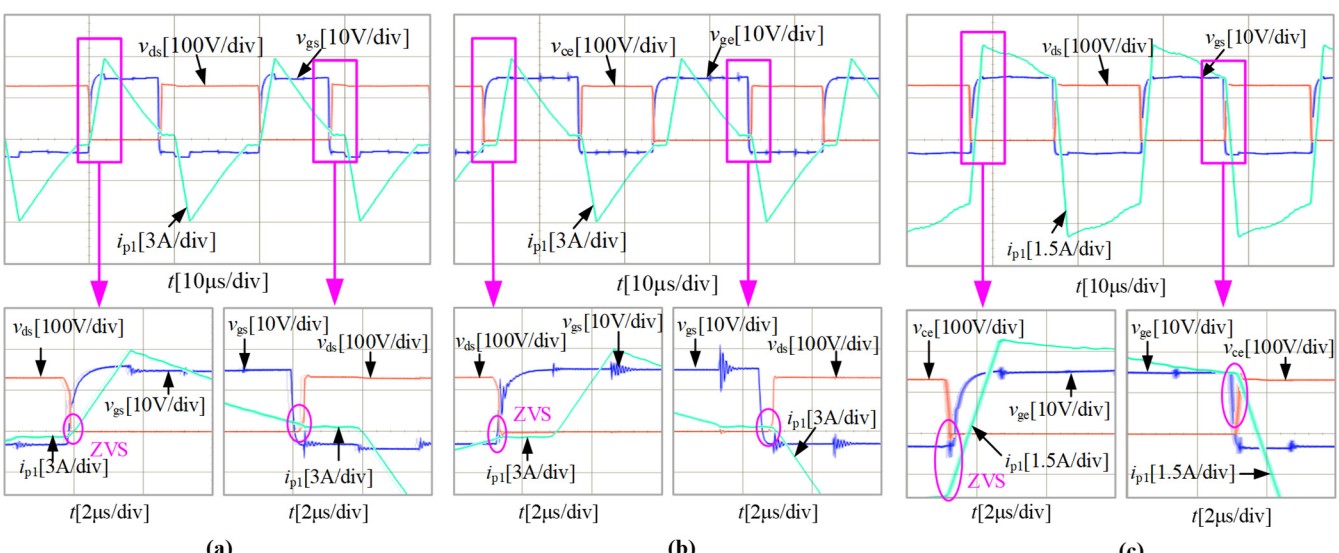

**Figure 17.** Switching characteristics of HSDC and ISOP-DAB with 405 V and 1000 W: (**a**) $S_{p11}$ of HSDC; (**b**) $S_{p14}$ of HSDC; (**c**) $S_{p11}$ of ISOP-DAB.

The switching characteristics for the primary voltage of 405 V and power of 2000 W are given in Figure 18. The primary switches of both HSDC and ISOP-DAB realize ZVS. The HSDC operates in Region 2, and the primary lower switch realizes zero-current switching (ZCS) with negligible turn-off losses.

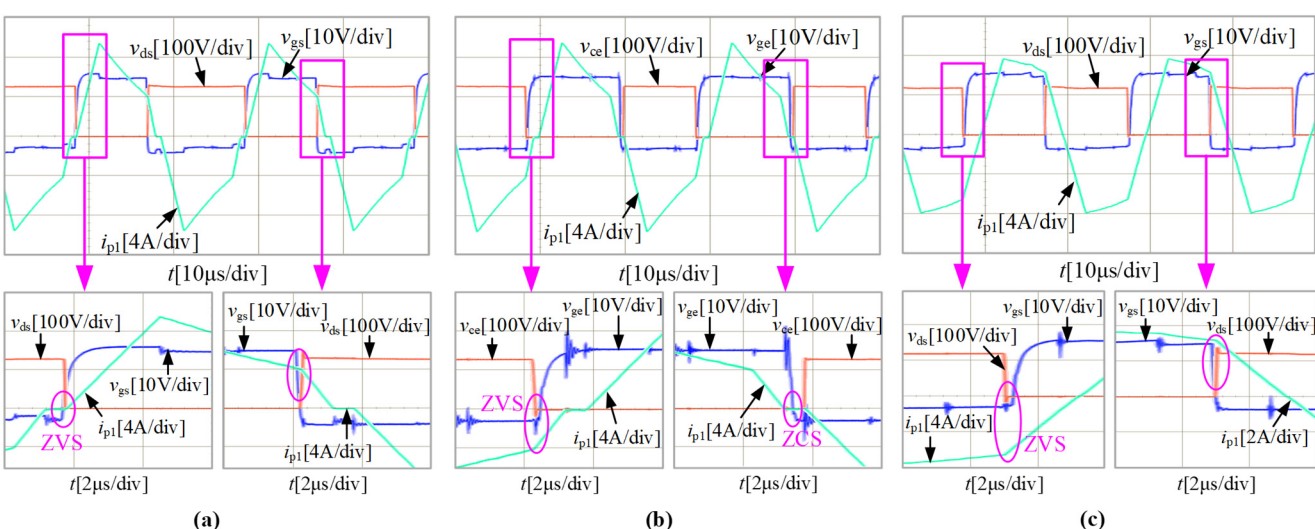

**Figure 18.** Switching characteristics of HSDC and ISOP-DAB with 405 V and 2000 W: (**a**) $S_{p11}$ of HSDC; (**b**) $S_{p14}$ of HSDC; (**c**) $S_{p11}$ of ISOP-DAB.

The switching characteristics of HSDC with the primary voltage 450 V and 495 V are shown in Figures 19 and 20, respectively. In the above conditions, the converter is operating in Region 2, and the switching characteristics of the primary switches are basically the same as those at 405 V and 2000 W. Therefore, the switching characteristics will not be analyzed in detail.

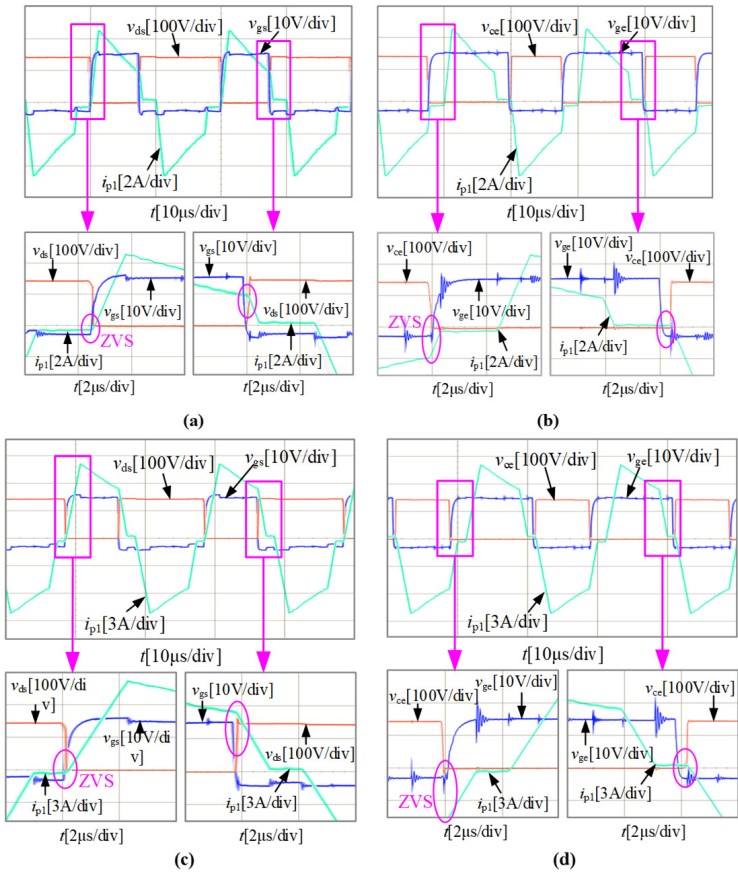

**Figure 19.** Switching characteristics of HSDC with 450 V: (**a**) $S_{p11}$ with 1000 W; (**b**) $S_{p14}$ with 1000 W; (**c**) $S_{p11}$ with 2000 W; (**d**) $S_{p14}$ with 2000 W.

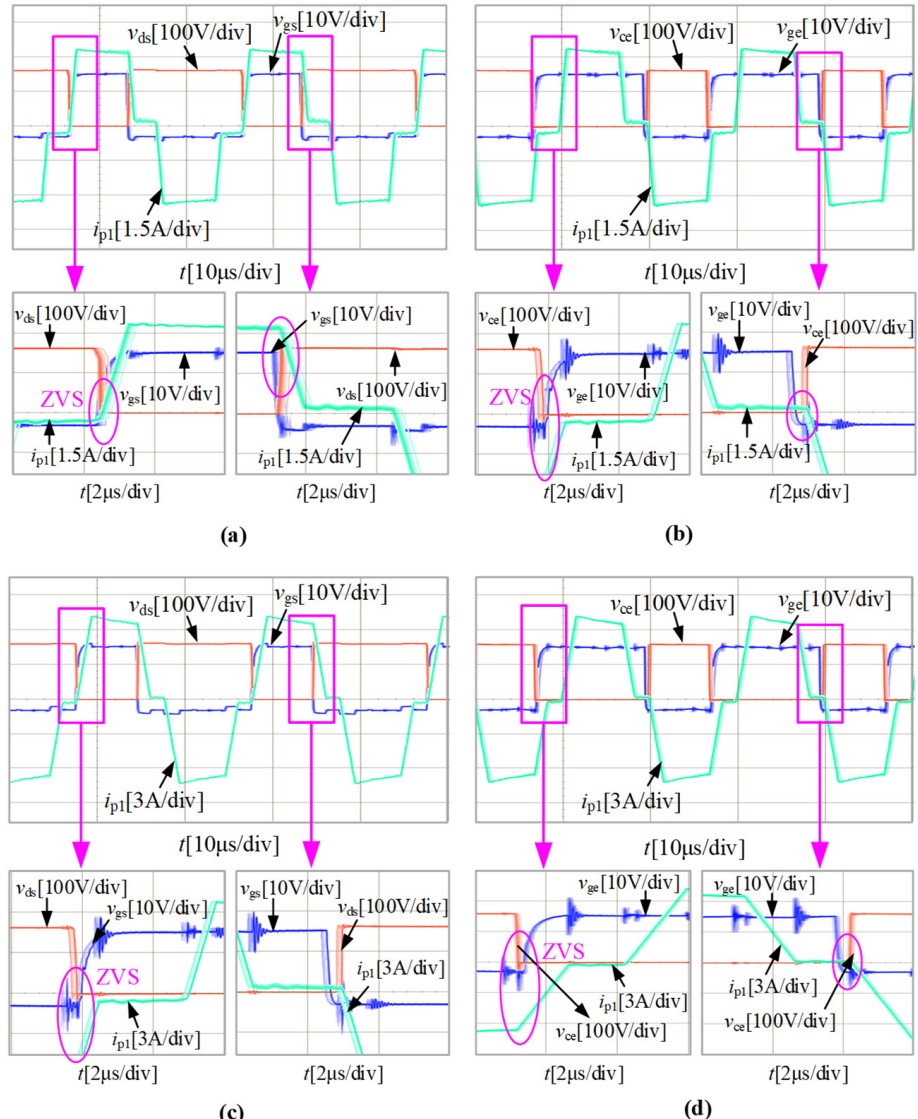

**Figure 20.** Switching characteristics of HSDC with 495 V: (**a**) $S_{p11}$ with 1000 W; (**b**) $S_{p14}$ with 1000 W; (**c**) $S_{p11}$ with 2000 W; (**d**) $S_{p14}$ with 2000 W.

The turn-off currents of the primary switches for the primary voltage of 405 V–495 V and power of 1000–2000 W are shown in Figure 21. The turn-off currents of upper switch for HSDC and ISOP-DAB are essentially the same when the primary voltage is 495 V. As the primary voltage drops, the turn-off current of HSDC will be lower than that of ISOP-DAB, and the difference between the two converters gradually increases. By comparing Figures 13 and 21, the trend and magnitude relationship of the turn-off current for the actual and theoretical results are basically consistent.

For HSDC and ISOP-DAB, the maximum voltages of the primary switches are both the capacitive voltages of the H-bridge where the switches are located, and the maximum voltages of the secondary switches are both the capacitive voltage of the secondary side. Therefore, the maximum voltages of the switches of both converters can be considered the same when the primary and secondary DC voltages are the same.

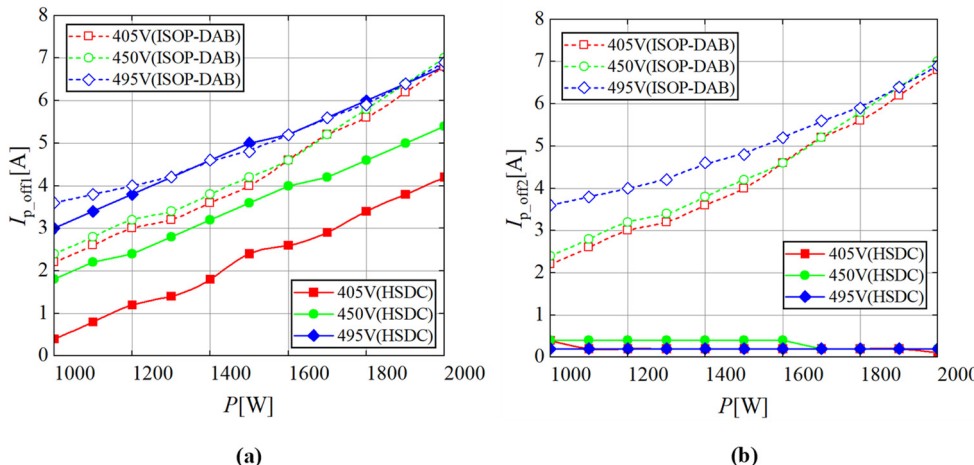

**Figure 21.** Turn-off current of primary switches of HSDC and ISOP-DAB: (**a**) Upper switches; (**b**) Lower switches.

The comparison of the current stresses of the switches for the HSDC and ISOP-DAB is shown in Figure 22, where $I_{p\_max}$ and $I_{s\_max}$ are the current stress of the primary and secondary switches, respectively. For the primary switches, the current stress of HSDC is higher than that of ISOP-DAB at lower primary voltage, and the difference between the two decreases as the primary voltage increases. When the primary voltage reaches 495 V, the current stress of HSDC is slightly less than that of ISOP-DAB. The magnitude relationship of the current stress of the secondary switches is similar to that of the primary switches, but the current stress of HSDC is still slightly higher than that of ISOP-DAB at the primary voltage of 495 V. In summary, the proposed OMS somewhat increases the current stresses compared to the existing strategy.

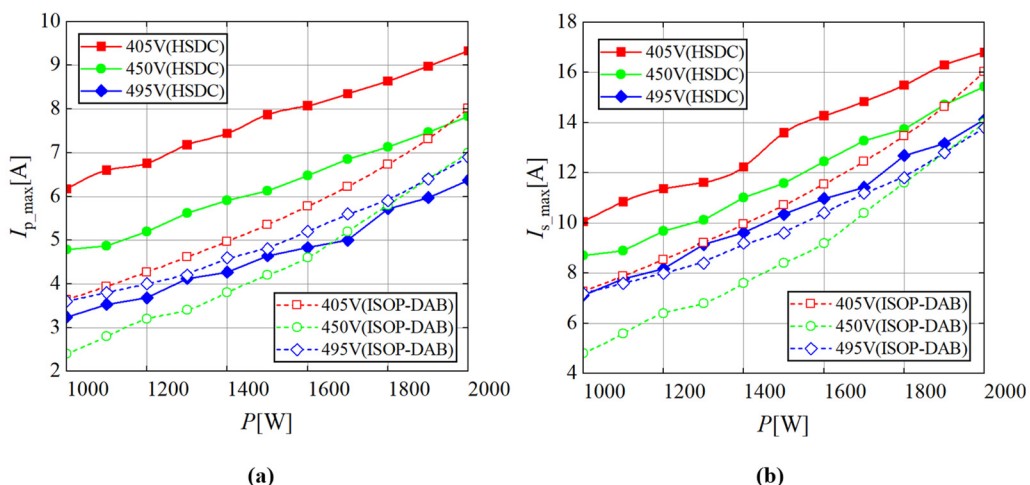

**Figure 22.** Comparison of the current stresses of the switches: (**a**) Primary switches; (**b**) Secondary switches.

Figure 23 illustrates the thermograms of switches at 450 V and 2000 W for 5 min of continuous operation, where the temperatures denote the average temperatures of the switches of the H-bridge or three-phase half-bridge bridge. The average temperatures for the switches of HSDC are all lower than those of ISOP-DAB, so the switching losses of HSDC are lower than those of ISOP-DAB, thus proving the effectiveness of OMS in reducing switching losses.

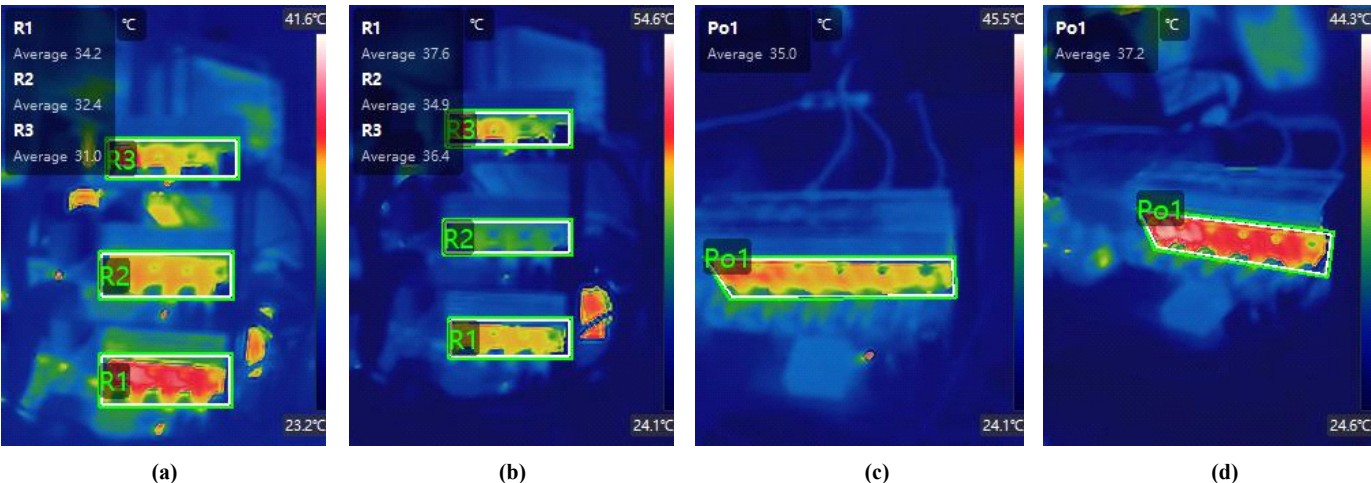

**Figure 23.** Thermogram of switches: (**a**) Primary switches of HSDC; (**b**) Primary switches of ISOP-DAB; (**c**) Secondary switches of HSDC; (**d**) Secondary switches of ISOP-DAB.

The efficiency of HSDC and ISOP-DAB is compared in Figure 24. When the power is in the range of 1000–1400 W, the efficiency of HSDC and ISOP-DAB is basically the same as the turn-off losses of HSDC and ISOP-DAB are not much different. As the power increases, the turn-off losses of the ISOP-DAB increase, and the efficiency gradually decreases. The turn-off currents for the primary upper switches of HSDC are lower than those of ISOP-DAB, and the lower switches realize low-current turn-off. As a result, the turn-off losses of HSDC are low and the efficiency remains basically the same compared to the low-power region. The optimization of the HSDC at full load is the most effective, and the difference in efficiency at full load under experimental condition ranges from 1.65% to 4.04%. When the primary voltage is 495 V, the difference between the turn-off currents of HSDC and ISOP-DAB is the smallest, and the difference between the efficiency is the smallest. When the primary voltage is 405 V, the difference between the turn-off currents of the two converters is the largest, and the difference between the efficiency is the largest as well. In summary, the proposed OMS can effectively reduce the losses in the high-power region and realize the efficient operation of the converter.

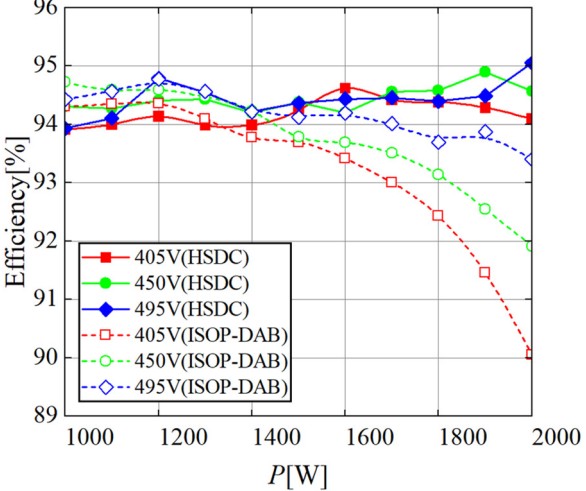

**Figure 24.** Efficiency of HSDC and ISOP-DAB.

## 6. Conclusions

In order to reduce the losses and costs of DCSST, this paper proposes a Si IGBT/SiC MOSFET hybrid isolated bidirectional DC–DC converter and an optimized modulation

strategy. The operating regions of the proposed converter under duty cycle modulation are divided by TDA method and each region is modeled. An optimization problem is established with the square of the rms value of the primary AC current as the optimized objective and the switching characteristics and the transmission power characteristics as the constraints. The optimization problem is solved by the augmented Lagrangian genetic algorithm (ALGA), and the OMS for the proposed converter is deduced. Finally, the effectiveness of OMS in reducing switching losses and improving efficiency is verified by comparison experiments with ISOP-DAB. The switching costs of the proposed converter are reduced by 27.3% and the efficiency is improved by up to 4.04% compared to the existing ISOP-DAB.

**Author Contributions:** Conceptualization, J.H. and Z.L.; methodology, J.H.; software, Y.W., Z.L., and H.Z.; formal analysis, J.H. and Y.W.; data curation, Z.L. and K.L.; writing—original draft preparation, J.H. and Y.W.; writing—review and editing, J.H. All authors have read and agreed to the published version of the manuscript.

**Funding:** This work was funded by National Natural Science Foundation of China (52130710).

**Data Availability Statement:** The data that support the findings of this study are available from the corresponding author upon reasonable request.

**Conflicts of Interest:** Author Zhenfeng Li was employed by the company Zhejiang Huayun Electric Power Engineering Design & Consultation Co., Ltd. The remaining authors declare that the research was conducted in the absence of any commercial or financial relationships that could be construed as a potential conflict of interest.

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
