# Peer review of "A Si IGBT/SiC MOSFET Hybrid Isolated Bidirectional DC–DC Converter for Reducing Losses and Costs of DC Solid State Transformer"

_electronics, doi:10.3390/electronics13040801_

Round 1

Reviewer 1 Report

Comments and Suggestions for Authors

Thank you for submitting your work. Having read through it, I believe there are a few main questions that need to be answered. First off, if the main purpose of the work is to improve loss and cost aspects of a dc-SST, how much improvement do we have as a conclusion of this work? Moreover, how is the proposed approach different from existing ones? These should be clarified and elaborated in the abstract/introduction/conclusion parts of the paper, since otherwise it is difficult for the readers to appreciate the main contributions of the work. In addition, please find below a few detailed comments:

1) What are the limitations of the proposed Si IGBT/SiC MOSFET hybrid isolated bidirectional DC-DC converter and optimized modulation strategy?

2) Comparison or benchmarking with existing DC solid state transformer (DCSST) technologies is missing. As such it difficult to assess the relative performance and limitations of the proposed approach.

3) The potential impact of the proposed converter on other system components or the overall system efficiency was not discussed. Furthermore, the scalability or adaptability of the proposed converter to different voltage levels or grid configurations was also not discussed.

4) In terms of hardware configuration (switches), comparative advantage of the proposed approach should be clarified with comparison to existing solutions. 

5) What is the impact of the OMS on other performance aspects of the DAB converter? e.g., voltage regulation and power quality.

6) What about the computational complexity or time required to implement the augmented Lagrangian genetic algorithm (ALGA) for solving the optimization problem associated with the OMS? This can help clarify the potential challenges or limitations of implementing the proposed OMS in practical applications.

Comments on the Quality of English Language

No major issues.

Reviewer 2 Report

Comments and Suggestions for Authors

Dear authors,

Please add the text and comments before the figure. Please pay attention at the name of the components from the figure, to not overlap with the component symbol(example, figure 2, name Sp13, Sp14, Sp22).

For a better understanding, please find a method to highlight the region on the figure 5.

Rows 216, 218, please pay attention at the symbol name, because it is different than the figure.  

Row 295. It missed a letter at kHz.

Figure 3 is missing.

Ensure that all references are up-to-date and relevant. Consider adding more recent studies to demonstrate the current relevance of your work, and try to highlight the importance of your work.

You simulate the circuit before to practically implemented?

What you use for control the switches?

Do you have a more comprehensive analysis of the comparison between the converters, where you take into account the maximum voltage and current stresses on your components?

Reviewer 3 Report

Comments and Suggestions for Authors

This paper suggests a methodology to reduce conduction loss in Si IGBT/SiC MOSFET Hybrid DC‐DC Converter (HSDC) using augmented Lagrangian genetic algorithm(ALGA) based on the time domain anlaysis for the parameters to be optimized. The authors addressed the fundamental switching algorithms used in the converter concretely as well as the calculated and experimental results including final efficiency comparison between the two convertes; HSDC and ISOP-DAB. This paper is recommended to be pusblished, however, after some revisions/answers to the reviewers' comments.

1. Line 124:   "...defined the lower switches" to "... defined as the lower switches"

2. Line 164:  "positive operating" could be "forward operating".  

3. equation 18:  x in "c_i(x)" could be replaced by large X with bold character.

4. Line 268:  It would be good if the authors include initial X0 values they used.

5. Fig. 10 is missing, and should be added.

6. Line 413: Make it sure that the plots in Fig. 21 come from experiment, not from theoretical ones.

Round 2

Reviewer 1 Report

Comments and Suggestions for Authors

The authors have addressed most concerns from the first round review. No further questions from the reviewer.

Reviewer 2 Report

Comments and Suggestions for Authors

Dear authors,

Congratulations for your work. The article look much better and professional. Just small typo mistakes you need to correct.

Comments on the Quality of English Language

Just small typo mistakes, that you need to correct through the paper.